# Lightweight Two-Layer Control Architecture for Human-Following Robot

**DOI:** 10.3390/s24237796

**Published:** 2024-12-05

**Authors:** Gustavo A. Acosta-Amaya, Deimer A. Miranda-Montoya, Jovani A. Jimenez-Builes

**Affiliations:** 1Instrumentation and Control Department, Faculty of Engineering, Politécnico Colombiano Jaime Isaza Cadavid, Medellín 050022, Colombia; gaacosta@elpoli.edu.co; 2Department of Computer and Decision Sciences, Faculty of Mines, Universidad Nacional de Colombia, Medellín 050034, Colombia; demiranda@unal.edu.co

**Keywords:** mobile robotics, computer vision, human-following robot, autonomous robot, RGB-D sensor, fuzzy logic control, behavior-based control architecture, embedded controller

## Abstract

(1) Background: Human detection and tracking are critical tasks for assistive autonomous robots, particularly in ensuring safe and efficient human–robot interaction in indoor environments. The increasing need for personal assistance among the elderly and people with disabilities has led to the development of innovative robotic systems. (2) Methods: This research presents a lightweight two-layer control architecture for a human-following robot, integrating a fuzzy behavior-based control system with low-level embedded controllers. The system uses an RGB-D sensor to capture distance and angular data, processed by a fuzzy controller to generate speed set-points for the robot’s motors. The low-level control layer was developed using pole placement and internal model control (IMC) methods. (3) Results: Experimental validation demonstrated that the proposed architecture enables the robot to follow a person in real time, maintaining the predefined following distance of 1.3 m in each of the five conducted trials. The IMC-based controller demonstrated superior performance compared to the pole placement controller across all evaluated metrics. (4) Conclusions: The proposed control architecture effectively addresses the challenges of human-following in indoor environments, offering a robust, real-time solution suitable for assistive robotics with limited computational resources. The system’s modularity and scalability make it a promising approach for future developments in personal assistance robotics.

## 1. Introduction

The aging population has become a matter of high priority for our society. Elderly people require innovative personal assistance solutions to maintain quality of life and the possibility of aging in their communities and homes [1,2]. Mobile robotics and computer vision are two of the most promising technologies for personal assistance in today’s homes [3,4,5]. More and more service robots are being equipped with vision systems that provide richer environmental information than laser scanners and ultrasound ranging systems [6,7].

A broad overview of the potential uses of service robots to assist elderly and disabled individuals is presented in [8]. The scientific literature addressing the application of robotic technologies for monitoring health, safety, mobility, and well-being, as well as assisting elderly individuals with daily living activities, is systematically reviewed, analyzed, and discussed in [9,10].

Innovations such as care robots are proposed in [11,12] to provide support to the daily life activities of elderly people. The robots are equipped with a laser range finder and a floor-sensing system for detecting both moving objects and people. A service trolley and a wheelchair are additionally equipped with inertial sensors to facilitate their localization inside the room. Additionally, a humanoid mobile robot provides services for the recognition and transport of objects [13]. This robot integrates an RGB-D sensor, which is utilized exclusively for object classification and manipulation tasks, rather than for tracking the movement of people. The authors placed this restriction because they considered that the monitoring of people’s activity with visual systems could become invasive to the privacy of individuals [14,15]. The main disadvantage of this facility is its high cost. In our approach, the use of an RGB-D sensor does not compromise the privacy of individuals since the captured frames are only used locally to extract distance and angular displacement information used by the fuzzy controller, and this information is neither stored nor transmitted through communication networks [16,17,18].

A relevant issue has to do with the acceptance or not by the users of the new technologies of assistance and care. In that sense, a study related to the answer to the question of how both older people and caregivers perceive the possibility of using an assistive robot for nutritional support was conducted in [19]. According to the authors, most participants considered the possibility of using a robot to improve the diet of the elderly by giving advice on healthy eating or controlling the amount of food consumed [20]. Another study conducted in [21] was focused on identifying the requirements and needs of people with Alzheimer’s and early dementia concerning robotic assistants.

One of the most required tasks in assistive robotics is the safe tracking of people. A human-following mobile robot system equipped with a laser range scanner was developed in [22]. The data provided for the laser were used to detect the position of the target based on the estimation of both shin positions of the person.

The authors consider the most significant contributions of this paper to be (1) providing evidence that a lightweight behavior-based control architecture enables a robotic system with limited computational resources to perform complex tasks, such as tracking people indoors; (2) offering a detailed explanation of the processes involved in modeling, design, testing, and adjusting each layer of the proposed architecture, and its integration into a final solution; and (3) presenting extensive validation experiments that demonstrate the feasibility of the proposed solution to address complex problems, such as the autonomous tracking of people by a robot.

This paper presents the design and implementation results of the human-following mobile robot platform Carlitos-Care. The article is structured as follows: Section 2 details the development of the mobile robot platform, including the effector modeling and the design and implementation of the embedded controllers. Section 3 presents the results and discusses the validation experiments, focusing on the behavior-based control architecture and the design and characterization of fundamental fuzzy behaviors. Finally, Section 4 summarizes the conclusions.

## 2. Materials and Methods

The development of the human-following mobile robot platform Carlitos-Care was guided by principles of modularity, scalability, simplicity, and cost-effectiveness. This section outlines the technical components and methodologies employed in constructing the robotic platform, including the modeling of effectors, the design and implementation of embedded controllers, and the integration of control systems. Each step in the design process was meticulously planned and executed to ensure the platform’s capability to perform complex tasks within limited computational constraints.

### 2.1. Mobile Robotic Platform

The mobile robot platform was developed having in mind modularity, scalability, simplicity, and low-cost criteria. It was named “Carlitos” for Car-Like Autonomous Robot System and has been used by the authors to research and teach engineering. The robotic platform is organized into layers: effectors, control, sensing, and communications. From a hardware perspective, each level corresponds to a printed circuit board (PCB) with a dodecahedron pattern. Each PCB not only contains the electronic components and connectors but also constitutes an integral part of the structure of the robot.

Figure 1 shows the block diagram and physical appearance of the follower robot. The layer of effectors is composed of two DC motors with gearboxes and quadrature encoders coupled to the axes of the motors. A dual H-bridge driver amplifies the PWM signals generated by the control layer. The main component of the control layer is a 32-bit NXP microcontroller, specifically MCF51QE128 (Freescale Semiconductor, Austin, EE. UU).

Other than odometry, the sensing layer has a CMPS03 digital compass for the estimation of the robot’s orientation and a sonar ring with twelve ultrasonic modules SRF02 for distance measurements of surrounding obstacles. Wireless data transfer between the robotic platform and a host PC is provided by an IEEE 802.5.4 IMS module. The system is powered by a couple of 7.4 V, 2200 mA-h, lithium-ion batteries (Li-ion).

### 2.2. Effector Modeling and Control

Algorithm 1 was implemented to facilitate the mathematical modeling of the robot’s effectors. It allows researchers to apply a time-variant PWM (Pulse-Width Modulation) signal to both DC motors, acquire data from rotary encoders, and transmit this information to a computer through an 802.15.4 wireless interface. A frequency of *f_s_* = 1/*T* = 10 Hz was established for data sampling.

The robot was placed in an obstacle-free indoor environment for the data acquisition experiments of the encoders coupled to DC motors, waiting for a start command from the host PC. Upon receipt of the order, a PWM signal with an initial DC (Duty Cycle) value of 20% was applied to each motor. Every 1.5 s, the DC value was increased by 20% until the maximum value of 80%. After reaching this value, the algorithm began to decrease this value in steps of 20% every 1.5 s until it reached again the minimum value of 20%; at that point, the experiment concluded, and the robot stopped. During robot displacement, the count of pulses from the encoders was sent to the host PC at the sampling rate previously mentioned.

Figure 2 shows the responses of both the left and right motor to the time-variant PWM signal. We use a First-Order Plus Dead-Time (FOPDT) model to describe the dynamics of the motors. With a proper choice of system delay *θ*, time-constant *τ*, and system gain *K* parameters in Equation (1), a useful model for the low-level controller’s design purposes was obtained.
(1)$$G\left(s\right)=\frac{Ke^{-\theta s}}{\tau s+1}=\frac{2.636e^{-0.1159s}}{0.2222s+1}$$

Specifically, step response data corresponding to a 40% to 50% applied PWM signal were used to develop the effector models. The System Identification Toolbox™ in Matlab^®^ (version R2021a, MathWorks, Portola Valley, CA, USA) facilitated this process, yielding a correlation factor of 94.02%. With this model in hand, the design of the control layer for the effectors was undertaken using both pole placement and internal model methods, which are described below.
**Algorithm 1**: Application of time-variant PWM signal to robot’s effectors. Source: Authors.Input: Start command from host PC through an 802.15.4 linkOutput: Pulse counts from encoders in host PC through ab 802.15.4 link Initialize internal hardware of microcontroller:  UART at 9600,8,N,1 for 802.15.4 communications; timer TPM3 for 1 KHz PWM signals generations;  TPM1, TPM2 for count pulses of encoders;  interruptions of internal RTC for 0.1 s sampling interval  *Flag_UpDown_* = 0; *DCLeft_motor_* = 0; *DCRight_motor_* = 0;  *CountPulses_rightmotor_* = 0; *CountPulses_leftmotor_* = 0;  repeat   while *uart reception register empty* do    wait;   end   *RcvCommand = uart reception register*;  until *RcvCommand ≠* “*s*”;  repeat   if *Flag_UpDown_* = 0 then    *DCLeft_motor_* += *PWM_UpDown_*;    DCRight_motor_ += *PWM_UpDown_*;   end   else
*   DCLeft_motor_* −= *PWM_UpDown_*;    *DCRight_motor_* −= *PWM_UpDown_*;   end   *PWM_timeinterval_* = 15;   repeat    while *no RTC interruption* do     *wait*;    end    *CountPulses_Leftmotor_* = *internal count register TPM1*;    *CountPulses_Leftmotor_* = *internal count register TPM1*;    while *uart busy* do     *wait*;    end;    *uart transmition register* = *CountPulses_Leftmotor_*;    while *uart busy* do     *wait*;    end    *uart transmition register* = *CountPulses_Rightmotor_*;    *CountPulses_Leftmotor_* = 0; *CountPulses_Rightmotor_* = 0;    *PWM_timeinterval_* ——;   until *PWM_timeinterval_* = 0;   if *DCLeft_motor_* = 80 *and DCRight_motor_* = 80 then    *Flag_Updown_* = 1;   enduntil *DCLeft_motor_* < 20 *and DCRight_motor_* < 20;

#### 2.2.1. Pole Placement Method

In this method, requirements on the closed-loop response are obtained by placing a pair of complex conjugate poles in the s-plane [23]. The next equation corresponds to the transfer function regardless of the delay.
(2)$$G\left(s\right)=\frac{11.8632}{s+16.3636}=\frac{0.725}{0.06111s+1}$$
where *τ_eq_* = 0.0611 is used to calculate the sampling period as follows:(3)$$0.2\left(\tau_{eq}+\theta \right)\leq T\leq 0.6\left(\tau_{eq}+\theta \right)$$
(4)0.0354≤T≤0.1062

A sampling period *T* = 0.1 s close to *θ* = 0.1159 was selected and implemented through periodic interrupts of the internal RTC (Real-Time Counter) module of the MCF51QE128 microcontroller. In the z-domain, the pulse transfer function of the system in an open loop is given by Equation (5).
(5)$$HG\left(z\right)=\frac{0.8306\left(z+0.1501\right)}{z^{2}\left(z-0.6376\right)}$$

Considering a *D*(*z*) PI controller, the characteristic equations will be
(6)$$1+D\left(z\right)HG\left(z\right)=0$$
(7)$$1+\frac{q_{0}z+q_{1}}{z-1}\cdot \frac{0.8306\left(z+0.1501\right)}{z^{2}\left(z-0.6376\right)}=0$$

Thereby, we obtain
(8)$$z^{4}-1.637z^{3}+\left(0.8306q_{0}+0.6376\right)z^{2}+0.124673\left(q_{0}+6.66223q_{1}\right)z+0.124673q_{1}=0$$

To meet the requirements of a bounded response in the 2% band, a settling time *t_s_* = 0.711 s, and a damping coefficient *ζ* = 0.8, we have
(9)$$\omega_{n}=\frac{4}{\zeta t_{s}}=7.032\frac{rad}{s}$$
where *ω_n_* is the natural frequency. The dominant poles must be placed in
(10)$$z_{1,2}=\left|z\right|e^{\pm \theta i}$$
(11)$$\left|z\right|=e^{-\zeta \omega_{n}T}=0.5697$$
(12)$$\theta =\frac{180}{\pi }\omega_{n}T\sqrt{1-\zeta^{2}}=24.176^{\circ }$$
(13)$$z_{1,2}=0.5697e^{\pm 24.176^{\circ }i}=0.5197\pm 0.2333i$$

The fourth-order characteristic polynomial that meets the specifications of the dynamic behavior of the system will then be
(14)$$\left(z-0.5197+0.2333i\right)\left(z-0.5197-0.2333i\right)\left(z+a\right)\left(z+b\right)=0$$
(15)$$z^{4}+\left(a+b-1.0394\right)z^{3}+\left(ab-1.0394a-1.0394b+0.324517\right)z^{2}+\left(0.324517a-1.0394ab+0.324517b\right)z+0.324517ab=0$$

Equating similar terms from Equations (1) and (8), we have
(16)−1.6376=a+b−1.0394
(17)$$0.8306q_{0}+0.6376=ab-1.0394a-1.0394b+0.324517$$
(18)$$0.124673q_{0}+0.8306q_{1}=0.324517a-1.0394ab+0.324517b$$
(19)$$0.124673q_{1}=0.324517ab$$

By solving the system of Equations (16)–(19), we can determine the values of *q*_0_ and *q*_1_
(20)$$q_{0}=0.2853,q_{1}=-0.1867$$

Finally, from these values we obtain the pole placement controller (PPC)
(21)$$D\left(z\right)=\frac{M\left(z\right)}{C\left(z\right)}=\frac{q_{0}z+q_{1}}{z-1}=\frac{0.2853\left(z-0.65439\right)}{z-1}$$

#### 2.2.2. Internal Model

This model-based design method states that control can be achieved when the control system incorporates, implicitly or explicitly, some representation of the process to be controlled [24,25,26]. The design of an internal model controller (IMC) begins with determining the parameter *λ*, an adjustable parameter that defines the response speed of the system, the controller gain *K_c_*, and the integral time *τ_i_* using
(22)$$\lambda \geq 1.7\theta ,\lambda \geq 1.7\left(0.1159\right),\lambda \geq 0.197$$
(23)$$\lambda \geq 0.2\tau ,\lambda \geq 0.2\left(0.2222\right),\lambda \geq 0.0444$$

For a low gain in the controller and a slow response of the feedback system, we choose *λ* = 0.197. The other two parameters can be calculated as follows
(24)$$K_{c}=\frac{\tau }{K\lambda }=\frac{0.2222}{\left(2.636\right)\left(0.197\right)}=0.4278$$
(25)$$\tau_{i}=\tau =0.2222$$

Equations (26) and (27) are used to calculate the *q*_0_ and *q*_1_ parameters.
(26)$$q_{0}=K_{c}\left[1+\frac{T}{2\tau_{i}}\right]=0.4278\left[1+\frac{0.1}{2\times 0.2222}\right]=0.5162$$
(27)$$q_{1}=-K_{c}\left[1-\frac{T}{2\tau_{i}}\right]=-0.4278\left[1-\frac{0.1}{2\times 0.2222}\right]=-0.3265$$

From these values we obtain IMC given by Equation (28).
(28)$$D\left(z\right)=\frac{M\left(z\right)}{C\left(z\right)}=\frac{q_{0}z+q_{1}}{z-1}=\frac{0.5162\left(z-0.6325\right)}{z-1}$$

The difference equation that implements the control law by the internal model method is Equation (29), where *m*(*k*) is the actual output, *m*(*k* − 1) the previous control output, and *e*(*k*) and *e*(*k* − 1) the actual and previous error signals.
(29)$$m\left(k\right)=q_{0}e\left(k\right)+q_{1}e\left(k-1\right)+m\left(k-1\right)=0.5162e\left(k\right)+0.3265e\left(k-1\right)+m\left(k-1\right)$$

#### 2.2.3. Low-Level Control Architecture

Figure 3 shows the proposed low-level architecture, where the controller (*z*) is selected based on the performance in the experimental tests (see Section 3.1).

### 2.3. Linear and Angular Displacements

Currently, the control paradigms that dominate the robotics scene are deliberative, reactive, and hybrid [27], each with distinct characteristics and computational requirements. Deliberative architectures excel in route planning and environment modeling through techniques such as SLAM and search algorithms like A* or D*. However, these approaches are computationally expensive due to the intensive processing needed to build and maintain maps and to manage uncertainties using probabilistic methods. Consequently, they often require advanced hardware, limiting their applicability in low-cost systems [28].

Hybrid architectures, on the other hand, combine deliberative planning with reactive layers that execute immediate actions in response to stimuli. While they offer a balance between long-term planning and real-time responsiveness, they inherit the high computational cost of environment modeling and global planning from deliberative systems. Additionally, the complexity of coordinating these layers increases their resource demands, making them more suitable for systems with intermediate computational capabilities but less ideal for resource-constrained hardware [29].

In contrast, reactive architectures based on fuzzy behaviors, such as the one proposed in this study, eliminate the need for environment modeling and global planning. By directly processing sensor data and employing fuzzy rules to manage uncertainty, these architectures achieve efficient control with simple mathematical operations and real-time responsiveness. This significantly reduces computational load, making them well suited for implementation on low-cost microcontrollers without compromising performance in specific applications, such as human-following tasks [30].

We chose a reactive behavior-based scheme for our high-level control layer because of the following advantages: (a) low computational cost, (b) easy implementation through soft computing methods, (c) high-speed response to changes in the environment thanks to the direct mapping between sensing and actuating, and (d) a high degree of modularity [31]. It is noteworthy that some of its disadvantages are the difficulties in establishing a set of adequate basic behaviors to support a given task of navigation, the emergence of unpredictable behaviors that occur from the process of switching behaviors according to the environment state, and no execution of planning tasks because it does not have a model of the environment [32]. Now, when we consider the problem of an autonomous robot that follows a person, a heuristic approach suggests that if the person is just right in front of the robot and walks away without altering their orientation angle to the robot, a linear displacement allows the robot to follow the person without losing them [33]. However, when the person makes sudden changes in orientation, the robot must quickly correct its orientation, so that the person will not disappear from the field of view (FOV) of the robot’s sensor [34]. Based on this reasoning, we identified two basic behaviors:

Linear Displacement. The objective of this behavior is to maintain an approximate distance of 1.5 m between the robot and the person whom it follows. This behavior generates a velocity output that is sent as references (SPright and SPleft) to the left and right embedded controllers (see Figure 4). With the same speed reference for both motors, approximately one linear displacement of the differential robot is achieved. To a greater distance, a greater reference speed is generated, and the laser distance has a lower reference value. This behavior takes control of the robot’s effectors when the person walks away, in the perpendicular direction to the sensor surface and a FOV delimited by *ϕ_min_* ≤ *ϕ* ≤ *ϕ_max_*, where *ϕ* is the angular position of the person with respect to the RGB-D sensor and *ϕ_min_* and *ϕ_max_* are the angular values that determine the threshold for the activation/deactivation of this behavior.

Angular Displacement. This behavior takes control of robot’s effectors when the angular position of the person is out of the range *ϕ_min_* ≤ *ϕ* ≤ *ϕ_max_* to the normal sensor surface (see Figure 4). The objective of this behavior is to keep the person within the field of view (FOV) of the sensor and avoid losing track of them. This behavior operates at a higher hierarchical level and overrides the linear displacement behavior when the angular measurement falls outside the specified range. We use the Kinect Software Development Kit SDK v. 10 and its skeletal tracking functionality to obtain and track the coordinates of skeleton joints. This functionality provides 3D coordinates of twenty skeletal joints; using Matlab, we isolated the coordinates of the skeletal joint named spine 1 and took it to be the person’s position. As shown in Figure 4, the lateral displacement x and depth z were used to calculate the distance d and orientation ϕ of the person by means of Equations (30) and (31).
(30)$$d=\sqrt{x^{2}+z^{2}}$$
(31)$$\phi =tan^{-1}\left(\frac{x}{z}\right)$$

Both behaviors were synthesized employing Mamdani-type Fuzzy Inference Systems (FISs) [35]. Table 1 specifies the input and output variables associated with the fuzzy behavior-based control.

Each linguistic variable was characterized as a tuple of the form:(32)$$\left\{X,T\left(x\right),U,G,M\right\}$$
where *X* is the name of the linguistic variable, *T*(*x*) the set of terms associated with the fuzzy sets to be defined in the universe of discourse *U*, *G* are the syntactic rules for establishing the terms *T*(*x*), and *M* is a semantic rule that associates to each term in *T*(*x*) a fuzzy set. Below, a detailed discussion of the design process of the behavior-based fuzzy controllers is presented.

#### 2.3.1. Linear Displacement Fuzzy Behavior (LDFB)

The universe of discourse *U* for the input variable *e_d_* was established using Equation (33), where *spd* is the distance set-point and *d* is the distance to the person.
(33)$$e_{d}=spd-d$$

Assuming that a comfortable distance of the robot to a person is approximately *spd* = 1.5 m, since the experimental operating range of our Kinect sensor goes from 0.8 m to 3.8 m, the universe of discourse *U* can be established. For the output variable *V_r_*, minimum and maximum values of velocity *V_rmin_* = 0 and *V_rmax_* = 220 *pp*/*T* were established. The complete characterization of both input variables, according to Equation (17), can be seen in Table 2. The fuzzy sets with which the universe of discourse *U* was partitioned allow us to establish how far from the Comfort Distance (ComDi) the person being followed is. Set *nde*, for example, makes it possible to determine that the person is at a much greater distance than the ComDi, while *nmde* indicates that they are somewhat further away from this value. On the other hand, *zde* is used to establish that the distance between the robot and the person is very close to ComDi and *pde* that the distance is less than this value; in this case, the robot is very close, and this could make the person uncomfortable.

As for the output variable *ω*, its universe V was partitioned into four sets, where *rls* places reference values very close to zero to IMCs, in such a way that the robot moves very slowly or stops completely. Similarly, sets *rslm*, *rshm*, and *rsh* set, respectively, low, medium, and high reference speeds to the controllers. Figure 5 graphically presents the input and output fuzzy sets for LDFB.

Because one of our design assumptions was to develop a lightweight control architecture, we chose the simplest membership functions like trapezoidal (*trapmf*) and triangular (*trimf*). The set of inference rules for the LDFB is as follows:

*R*_1_: if *e_d_* is *nde*, then *ω* is *rsh.*

*R*_2_: if *e_d_* is *nmde*, then *ω* is *rshm.*

*R*_3_: if *e_d_* is *zde*, then *ω* is *rslm.*

*R*_4_: if *e_d_* is *pde*, then *ω* is *rsl.*

#### 2.3.2. Angular Displacement Fuzzy Behavior (ADFB)

Regarding this behavior, it is desirable for the robot to keep the person it follows directly in front. For a differential robot like the one in Figure 6, this means executing a pure angular displacement around its central point, **C_p_**, aiming to quickly respond to changes in the orientation of the person being followed so that they remain within the sensor’s field of view. In such a case, the motor speeds should be equal in both magnitude and direction of rotation.

Indeed, a counterclockwise rotation of the robot (CCWrobot) is achieved when the speeds of both motors are equal and both rotate in a clockwise direction (CW), as indicated in Table 3. This should be the robot’s movement when the person being followed moves to their left. On the other hand, when the person moves to the right, both motors should rotate counterclockwise (CCW) to produce a clockwise movement of the robot (CWrobot), as shown in Figure 6 and Table 3.

Based on the previous discussion, the sensor’s field of view, defined by −28° ≤ *ϕ* ≤ 28°, can be restricted to 0° ≤ *ϕ* ≤ 28°, using the sign of the original range to ensure that the embedded system generates the appropriate motor rotation directions. Since the goal is for the robot to maintain an orientation close to zero degrees relative to the person it is following, the universe of discourse for the variable of interest can be calculated as:(34)$$e_{a}=spa-\phi \Rightarrow e_{amin}=0^{\circ }-28^{\circ }=-28^{\circ },e_{amax}=0^{\circ }-0^{\circ }=0^{\circ },$$

The universe of discourse of this variable is then:(35)$$W=\left[e_{a}\in R|-28\leq e_{a}\leq 0\right]$$

For the linguistic variable *e_a_*, we considered the next set of terms:(36)$$T\left(x\right)=\{eanh,eanm,eanl,eaze\}$$
where:

*eanh*: Error angle negative-high. This indicates that the person that the robot is following is far away from the angular position set-point (*spa* = 0°).

*eanm*: Error angle negative-medium. The angular distance between the robot and the person is moderate.

*eanl*: Error angle negative-low. This indicates that the angular distance between the robot and the person is low.

*eaze*: Error angle zero. This means that the angular distance between the robot and the person is near zero, which is the desired angular distance.

We considered the following partition of the universe of discourse in fuzzy sets:

error angle negative-high eanh → trapmf [−28, −28, −15.88, −13.4].

error angle negative-medium *eanm* → *trimf* [−18.3, −12.24, −8.41].

error angle negative-low *eanl* → *trimf* [−11.9, −6.073, −1.77].

error angle zero *eaze* → *trimf* [−5.71, 0, 0].

We consider the reference velocity for angular displacement, *V_ra_*, like the output linguistic variable. The reference velocities of both the right and left motor controllers are equal in magnitude and have the same rotation direction such that the robot rotates on its middle point. Referring to Figure 6, if both motors rotate counterclockwise (*CCW*), the robot moves counterclockwise (*CW_robot_*). But, if the two motors rotate in the CW direction, then the robot moves clockwise around Cp. For this variable, we define the universe of discourse as (see Equation (37)):(37)$$Y=\left[V_{ra}\in R|0\leq V_{ra}\leq 100\right]$$
where *V_ra_* is given in *pp*/*T*. The expression *reference speed* and linguistic modifiers like *zero*, *positive low*, *positive medium*, and *positive high* are considered as syntactic rules. So, the set of linguistic terms for this variable are as follows (see Equation (38)):(38)$$T\left(x\right)=\{rsze,rspl,rspm,rsph\}$$
where:

*rsze*: Reference speed zero. This indicates that the reference velocity for both motors is too close to zero and the robot is stopped.

*rspl*: Reference speed positive low. Reference velocity is low. The robot makes angular displacements at low velocity.

*rspm*: Reference speed positive medium. The angular displacements of the robot are made at a moderate velocity.

*rsph*: Reference positive speed high. The robot makes fast angular displacements, avoiding that the person that it follows escapes from its visual field. The following fuzzy sets were considered in the implementation of this fuzzy controller:

reference speed zero *rsze* → *trimf* [0, 0, 1.603].

reference speed positive-low *rspl* → *trimf* [0.748, 3.95, 8.013].

reference speed positive-medium *rspm* → *trimf* [5.02, 15.92, 30.9].

reference speed positive-high *rsph* → *trapmf* [20, 22.33, 100, 100].

To implement the angular displacement behavior, the following rule base was established:

*R*_1_ if *e_a_* is *eanh*, then *V_ra_* is *rsph*.

*R*_2_ if *e_a_* is *eanm*, then *V_ra_* is *rspm*.

*R*_3_ if *e_a_* is *eanl*, then *V_ra_* is *rspl*.

*R*_4_ if *e_a_* is *eaze*, then *V_ra_* is *rsze*.

Table 4 summarizes and organizes this information for better identification, and Figure 5 graphically presents the input and output fuzzy sets for ADFB.

#### 2.3.3. Control Architecture

Figure 7 illustrates the two-layer control architecture proposed in this work, which integrates fuzzy behavior-based control (left) and low-level motor control (right). For our fuzzy behavior-based control architecture, the simplest coordination scheme was considered, named Arbitration or Action Selection. In this method, typically found in subsumption-type architectures, only one behavior can be active at one time [36]. In our case, the highest priority level was assigned to the angular displacement behavior, which ensures that the person followed by the robot does not disappear from the visual field of the Kinect sensor. Angular displacement behavior subsumes or cancels the linear displacement when the angular position of the person is in the range |*ϕ*| > 8°. The overall block diagram of our architecture is shown in Figure 7. The behavior selection module (BSM) implements the hierarchical coordination scheme based on the environment state captured with the Kinect sensor.

The outputs of the fuzzy behavior-based subsystem are the speed set-points *Vr*, which are sent to the robot via radio link 802.15.4 and are used as reference inputs for the PI-embedded controllers in the robot. Algorithms 2 and 3 implement the control architecture proposed. The first was coded in MatLab version R2021a and the second in C++ language version 20.
**Algorithm 2**: Behavior-based Control Layer. Source: Authors.Input: Time *t _f_* during which the person is followed by the robotOutput: Velocity set-point *V_r_* for PI embedded controllers, distance *d* between the person and the robot, angular position *ϕ* of the person respect to the robot, flag of active behavior *behave_flag*, and text file for data logging *data.txt**// behave_flag* = 1: robot angular displacement in *CW* //// *behave_flag* = 2: robot angular displacement in *CCW* //// *behave_flag* = 3: linear forward displacement in *CCW* //*V_r_* = 0; *t* = 0; *d* = 0; *ϕ* = 0;*behave_flag* = 0; *x* = 0; *z* = 0;Initialize the PC UART at 9600,8,N,1 for 802.15.4 communications;Open(“*data.txt*”, Write); /* create and open a data logging text file for write */Run the Kinect skeletal tracking functionality;repeat  *z* ← the *z* coordinate of the joint *spin 1*; *x* ← the *x* coordinate of the joint *spin 1*;  *d* = *sqrt*(*x^2^* + *y^2^*); *ϕ = tan*^−1^(*x*/*z*);  if −25 ≤ *ϕ* ≤ −8 or 8 ≤ *ϕ* ≤ 25 then   Deactivate *linear displacement behavior*;   *V_r_* = *fuzzy_angular_behavior*(*ϕ*);   if −25 ≤ *ϕ* ≤ −8 then    *behave_flag* = 2;   else    *behave_flag* = 1;   end  else   Desactivate *fuzzy angular displacement behavior*;   *V_r_* = *fuzzy_linear_behavior*(*d*);   *behave_flag* = 3;  end  *UART_transmission*(*behave_flag*,*V_r_*);  Write(“*data.txt*”, *V_r_*, *d*, *ϕ, behave_flag*);until *t* > *t _f_*;*behave_flag* = 0;*UART_transmission*(*behave_flag*);*UART_close*();Close(“*data.txt*”);

**Algorithm 3**: Embedded Control Layer. Source: Authors.Input: Active behavior flag *behave_flag*, Velocity set-point *V_r_*, and *stop_character* for stop the robotOutput: *DC_Left* and, *DC_Right* for setting the Duty Cycle of the PWM control signals for left and right motorsInitialize internal *UART* of microcontroller at 9600,8,N,1 for 802.15.4 communications;Enable interrupts for data reception in the *UART*;Configure the internal timer module *TPM1* to counting pulses of the left motor encoder;Configure the internal timer module *TPM2* to counting pulses of the right motor encoder;Configure the internal timer module *TPM3* to generate 1 *KH_z_* PWM control signals;Configure the internal *RTC* module to generate periodic interrupts each *T* = 0.1 s;*V_r_* = 0; *DC_left* = 0; *DC_right* = 0;*stop_character* = 0; *behave_flag* = 0;*LW_pulses* = 0; *RW_pulses* = 0;repeat  wait;until *t* > *t _f_*;// Interrupt Service Routine for internal *RTC* time out ISR *RTC_time_out*{*LW_pulses* ← (*TPM*1 *counting register*) // pulse counting left motor*RW_pulses* ← (*TPM*2 *counting register*) // pulse counting right motorswitch *behave_flag* do  case 1 do   *left motor rotation* ← *CCW*;   *right motor rotation* ← *CCW*; // robot rotates in CW  case 2 do   *left motor rotation* ← *CW*;   *right motor rotation* ← *CW*; // robot rotates in CCW  case 3 do   *left motor rotation* ← *CCW*;   *right motor rotation* ← *CW*; // robot moves forward  otherwise do *V_r_* = 0; end*DC_left = left_motor_PI_controller(LW_pulses, V_r_);**DC_right = right_motor_PI_controller(RW_pulses, V_r_);**TPM*1 *duty cycle register* = *DC_left*;*TPM2 duty cycle register = DC_right;**}*// Interrupt Service Routune for *UART* receptionISR *UART_reception*{*behave_flag* ← (*UART_reception_register*);*V_r_* ← (UART_reception_register);}

### 2.4. Experimental Designs

Experiments were designed to validate each layer of the control architecture, both at low and high levels. For the low-level control, performance metrics commonly used in classical control systems were considered. For the high-level control, the focus was on tracking the linear and angular displacements of a person relative to the robot. To evaluate behavior switching, test paths were designed that combined both linear and angular displacements. All experiments were conducted in controlled indoor environments, with no obstacles present, ensuring that the results were obtained under ideal conditions.

#### 2.4.1. Experimental Design for Embedded Controllers

Both control strategies, pole placement and IMC, were encoded and flashed in the internal memory of the microcontroller. Two simple paths of robot displacement were considered: linear and circular trajectories.

Approximate linear displacements were achieved when the speed set-points for both the left and right motor controllers had the same value. Time-based displacements of 15 and 20 s were regarded with velocity set-points of 100 and 150 *pp/T* (pulses per sampled period). A board marker was used to mark the path of the robot on the floor. For the performance evaluation of the controllers, the Integral of Absolute Error (IAE) and the Integral of Squared Error (ISE) were used. Equations (39) and (40) define these metrics.
(39)$$IAE=\int_{t_{0}}^{t_{f}}\left|e\left(t\right)\right|dt\approx \sum_{k=k_{0}}^{k_{f}}\left|e\left(kT\right)\right|$$
(40)$$ISE=\int_{t_{0}}^{t_{f}}e{\left(t\right)}^{2}dt\approx \sum_{k=k_{0}}^{kf}e{\left(kT\right)}^{2}$$
where *t*_0_ and *k*_0_ represent the initial time values, while *t_f_* and *k_f_* correspond to the final time values for the evaluation of these error-based metrics. Additionally, *k**T* denotes the sampling rate.

#### 2.4.2. Experimental Design for Fuzzy Behaviors

Our fuzzy behavior-based control architecture was validated through three different kinds of experiments: (a) linear approximation and tracking of a person, (b) angular approximation and tracking of a person, and (c) a combination of angular and linear displacements to test the behavior coordination mechanism proposed. These experiments were conducted in controlled indoor environments with no obstacles between the depth camera and the person being followed. As a limitation, the system’s performance has not yet been evaluated in dynamic or cluttered environments, where obstacles or occlusions could impact the accuracy of the depth sensing and the behavior coordination. This highlights the need for future studies to assess its robustness in more complex scenarios.

#### 2.4.3. Experimental Design for Linear Tracking

Figure 8 shows the first experimental set-up for linear displacement. A person who is just in front of the robot, at 3.5 m, remains standing. Once the robot is ordered to start moving, and after the Kinect recognizes and starts monitoring the skeletal joints, the linear displacement behavior begins to generate references of velocity that are sent to the embedded controllers to allow the robot to follow the person.

A variation in this experiment is shown in Figure 9. The person is initially located at 2.3 m from the robot P1. In this position, the person waits until the robot is close to 1.3 m in *P*_0_ and stops its movement. Then, the person begins to move away from the robot until they reach the final position _P2_ located at 3.2 m.

#### 2.4.4. Experimental Design for Angular Tracking

Figure 10 shows the first experimental set-up for the validation of angular displacement behavior. Initially, the person is located at point *A*, at an angular position of 25° with respect to the coordinate system of the Kinect^®^ sensor, at 2.5 m. The initial orientation of the robot coincides with the bisector *O* − *B*, which corresponds to 0° in the Kinect^®^ coordinate system. When the test begins, the robot detects that the person is located on its left side, so it makes a counterclockwise angular displacement until the angle of orientation of the person, relative to the sensor, is about 0°.

Once the person observes that the robot is in front of them, they begin a displacement over the path described by the arc *A* − *C* and stop in *C*, waiting for the robot to finish its rotation.

#### 2.4.5. Experimental Design for Behavior Coordination

The experimental set-up shown in Figure 11 corresponds to the first test of the coordination mechanism. Initially, the person is located at 2.5 m away, with an orientation angle of 25 relative to the robot. The robot is oriented in the direction of the bisector *O* − *B*. When the test starts, the person waits until the robot corrects its orientation and stops. From that moment, the person moves in the path comprising the arc *A* − *C* and, at point *C*, the person stops and waits a little while until the robot changes its orientation towards them. Then, the person starts moving on a straight line between points *C* and *D* and stops when they reach this latter point. The robot keeps moving until it finally corrects its distance and angular position with respect to the person who is waiting at point *D*.

Figure 12 describes the second conducted test for the validation of the coordination scheme of behaviors. Initially, the person is located 2 m away, with an orientation of −15° degrees relative to the robot, which is oriented in the direction of the line segment *O* − *B*. Once the test begins, the person waits until the robot is oriented toward them, and then the person starts moving over the circumference of radius r = 2.0 m in the CW direction. When the person reaches point *B*, they start moving in the direction of the straight line *B* − *C* and finally stop at point *C*. The test ends when the robot stops at about 1.3 m and inside of the range −8° ≤ ϕ ≤ 8°. Figure 13 shows the obtained results.

## 3. Results and Discussion

### 3.1. Results for Embedded Controllers

Figure 14 shows the response obtained for a linear distance of 458 cm and a time slot of 20.49 s with PPC, and Figure 15 shows the response of IMC. With the latter controller, 466 cm was reached in 20.63 s. In both cases, the velocity set-point was 100 *pp*/*T*. In both cases, it is seen in the figures that the steady-state error converges to zero. In addition, it was determined from the true trajectory of the robot on the floor that the maximum deviations to the reference straight line were 9 mm for PPC and 5 mm for IMC.

Figure 16 and Figure 17 show the results obtained for PPC and IMC for a set-point of 150 *pp*/*T*. With PPC, 514 cm was traveled in 15.43 s. In the case of IMC, the distance reached was 531 cm in 15.76 s. In the first case, the settling time (*t*_s_) was 1.5 s, and the steady-state error was zero. The maximum linear deviation from the straight line was 6 mm. In the latter, *t*_s_ was 1.2 s, the steady-state error was zero, no overshoot occurred, and the maximum linear deviation obtained was 12 mm.

Table 5 shows the values of IAE and ISE performance indexes obtained from linear displacement experiments with a set-point of 150 *pp*/*T*, for both the right and left motors of the controllers. The better controller is the one with the minimum IAE and ISE values. So, IMC was selected for the final implementation on the robot.

### 3.2. Results for Fuzzy Behaviors

The following are the results obtained for the validation of the fuzzy behaviors described in Section 2.3.1 and Section 2.3.2, corresponding to Figure 8, Figure 9 and Figure 10.

#### 3.2.1. Results for Linear Tracking

For the linear displacement shown in Figure 8, the robot reduces the distance to the person, and its linear velocity decreases until it eventually stops. Figure 18 shows how the initial distance of 3.5 m is reduced until it reaches the final value of 1.3 m. In this figure, and all the following ones, the first sampling periods (fifteen in this case) are used by the Kinect sensor to start the execution of the skeleton tracking algorithm. Figure 19 shows an initial value of the distance error of −2.3 m, and a final value of 0.2 m at the end of the path; this is the steady-state error of this fuzzy controller.

In Figure 20, the changes in set-points of velocity that are sent to the embedded controller can be seen. An initial set-point of 140 *pp*/*T* is generated for a quick approach to the person. Subsequently, velocity set-points are reduced to 60 *pp*/*T* s near *t* = 45 *T* s, and then to 20 *pp*/*T* s around *t* = 75 *T* s. Finally, when the robot reaches 1.3 m, this behavior sends a velocity set-point of 0 *pp*/*T* to the embedded controllers, and the robot stops.

As mentioned earlier, a variation in the experiment described in Figure 8 is shown in Figure 9. It can be seen how the robot approaches the person and stops at approximately 1.3 m. Soon after, the person begins to move away from the robot, and the fuzzy behavior responds by generating velocity references that allow the robot to follow the person to try to keep 1.3 m between them. This can be seen in Figure 21 around *t* = 100 *T* s. Once the person begins to move away from the robot, it is seen how the distance error increases its value up to −1.8 m (see Figure 22), just when the person reaches the *final position P_2_* around *t* = 125 *T* s.

Figure 23 shows how the behavior responds in this experiment. When it is detected that the person begins to move away, new velocity set-points are sent to the controllers, until they reach a value of 140 *pp*/*T* near *t* = 110 *T* s. This way the robot remains near the person, reacting to their displacement changes. As the robot approaches, the fuzzy behavior decreases the values of velocity set-points until the distance to the person, at the end of the path, is about 1.3 m (see Figure 21 and Figure 22). From that moment, the speed set-points take a value of 0.0 *pp*/*T*.

#### 3.2.2. Results for Angular Tracking

Regarding the angular displacement described in Figure 11, as soon as the robot detects that the person is located at 25° (Figure 24), it adjusts its position accordingly to maintain alignment with the person. The error of −25° (Figure 25) is processed by the angular displacement fuzzy behavior, that generates velocity set-points of 60 *pp/T* (see Figure 26) for both embedded controllers, but with a direction of rotation established according to the third entry of Table 4 input. The final position of 0° degrees is achieved at the end of t = 30 *T s*. The reference velocities are reduced until they reach a value near zero, when the robot is oriented toward the person.

This value is maintained for a period between *t* = 16 *T* s and *t* = 29 *T* s. The displacement of person along the arc *A* − *C* takes place during the period between 30 *T* and 75 *T*. Figure 24 shows that, in this lapse of time, the angular measurement deviates from the desired value of 0°. In response to this event, new velocity set-points, between 5 and 15 *pp*/*T*, are generated by the angular displacement behavior (Figure 26).

Figure 24 and Figure 25 show that from *t* = 75 *T*, the measured angle and the angular position error take a value near 0°. At this point, the person has completed their displacement, stopping at point *C*. This is also verified by observing Figure 26; the velocity set-points generated by this behavior are near zero from *t* = 75 *T*. Since the robot only carried out an angular displacement about its midpoint, the distance to the person did not change significantly, as is shown in Figure 27. Similar results were obtained when the person was moving from right to left, as seen in Figure 28.

### 3.3. Results for Behavior Coordination

In the test described by Figure 28, the person initially follows an arc, and then proceeds to make a linear displacement. Figure 29 and Figure 30 show the obtained results of this test. The initial measures of distance and orientation to the person are 2.75 m and 24°, respectively, which are taken at the first sampling periods shown in the same figures. These initial measurements are consistent with the experimental set-up of Figure 29. The behavior-based controller responds to this initial state, by activating the behavior of angular displacement in the CCW direction, which can be seen in Figure 30 marked with the label “2”. The controller sends the same speed set-points (60 *pp*/*T*) to both DC motors and commands them to rotate in the CW direction, so that the robot moves in the CCW direction (see Figure 6 and Table 3). After correcting the orientation of the robot and with the person located within the visual field bounded by −8° ≤ *ϕ* ≤ 8°, it is necessary to correct the distance, for which the behavior of linear displacement is activated up to about *t* = 90 *T* (label “3” in Figure 30). At this point, the robot has corrected its location relative to the person, who has remained motionless at point A. Then, the person begins their displacement along the arc *A* − *C*, and the fuzzy controller responds to this with multiple switching events between behaviors “1” (angular displacement in the CW direction) and “3” (linear forward displacement), the first of which is activated for longer. This switching is maintained between *t* = 90 *T* s and *t* = 120 *T* s, allowing the robot to follow the person without losing them.

At *t* = 120 *T* s, the person is already at point *C* and initiates their route along the line segment *C* − *D*. Figure 30 shows how from this moment, the behavior that remains most active is “3” (linear displacement), with sporadic activations of “1” (angular displacement in the CW direction) and “2” (angular displacement in the CCW direction) behaviors. Thus, it ensures that the person remains “visible” to the robot.

In the second behavior coordination test described in Figure 12, the person traces a full circle and then moves away linearly. The initial values of angular position and distance, measured with the Kinect sensor, are ϕ = 11.2° and d = 2.1 m, respectively; this is shown in Figure 31a and Figure 13b. During the movement of the person along the circumference, between points *A* and *B*, no significant changes are seen in the distance (Figure 29b), but changes were repeatedly seen in the angular position (Figure 29a). The circular displacement of the person occurs within 0 ≤ *t* ≤ 100 *T*, a period during which repetitive changes are observed in speed set-points (Figure 30a,b) and permanent switching between the behaviors of angular displacement in the CW direction and linear displacement occurs to correct the orientation and distance of the robot (Figure 30b). This happens until the person reaches point *B*, from which starts a linear displacement over the straight line *B* − *C*.

Once the person begins to move away from the robot, the behavior of linear displacement takes control of the robot’s effectors and sends a set-point of speed of about 138 *pp*/*T* within 120 *T* ≤ *t* ≤ 150 *T*.

The value of these set-points decreases as the robot reduces its distance from the person, until it reaches zero for *t* ≥ 270 *T*, as is shown in Figure 29b. As can be seen in Figure 31a, the behavior that remains most active for *t* ≥ 100 *T* is the linear displacement “3”, which it is consistent with the displacement of the person between points *B* and *C*.

Finally, it is seen in Figure 29d that after 270 *T*, the distance error is about 0.25 m, with which the robot approaches the person and stops at a near distance of 1.25 m. The final angular orientation error is near 4.5°, as shown in Figure 29c.

According to the results shown, the proposed architecture provides a balance between efficiency and robustness. By focusing on depth data and adaptable fuzzy rules, the system is ideal for low-cost platforms with limited computational resources. In comparison, alternative methods often involve complex techniques. For instance, feature fusion (e.g., facial recognition and clothing texture), as explored in [37,38], improves precision but demands high computational resources. Other methods rely on wearable devices like infrared LEDs or ultrasonic transmitters, which increase costs and limit flexibility. Deep learning methods, such as SSD (Single Shot MultiBox Detector, a deep learning method designed for real-time object detection in images), ensure robust detection but require specialized hardware like GPUs, elevating system complexity. In contrast, our proposed architecture stands out for its simplicity and efficiency, maintaining precise, adaptable control without requiring advanced hardware or intensive computations. This practical approach is well suited for applications constrained by economic and computational limitations.

## 4. Conclusions

This research presented a comprehensive approach to the design and implementation of a lightweight two-layer control architecture for a human-following robot, optimized for environments with limited computational resources. By combining a fuzzy behavior-based control system with low-level embedded controllers, the robot effectively tracks a human subject in real time while maintaining a safe distance of approximately 1.3 m. Although the skeletal tracking functionality of the Kinect sensor allows for the recovery of the human target in occlusion conditions, such occlusions were not incorporated in the experiments conducted in this work, which could be considered as future work. The system leverages an RGB-D sensor for precise distance and angular measurements, allowing for seamless human–robot interaction in indoor environments.

Experimental validation confirmed the robustness and effectiveness of the proposed architecture. The internal model control (IMC) method, employed in the low-level control layer, demonstrated superior performance. For the left and right motors, IMC achieved an Integral of Absolute Error (IAE) of 303 and 301, respectively, compared to 553 and 549 obtained with the pole placement method. Similarly, the Integral of Squared Error (ISE) values for IMC were 31,561 and 31,443 for the left and right motors, respectively, while pole placement yielded 48,955 and 47,855. These metrics indicate that IMC offers significantly better performance, making it the preferred choice for the low-level control layer of the proposed architecture. The fuzzy behavior-based control at the high level ensured fast response times and the flexible handling of complex navigation tasks, such as adjusting for sudden changes in human orientation. These results emphasize the system’s capability to provide reliable human-following behavior, making it well suited for assistive robotic applications where real-time responsiveness and safety are critical. In the linear displacement experiments, the robot successfully followed the person until reaching the safety and comfort distance of approximately 1.3 m (Figure 8 and Figure 18). For angular displacement, the robot achieved an orientation error of less than 1° relative to the person being followed during the movement, as shown in Figure 24 and Figure 25. This result is highly acceptable, as it keeps the person centered within the depth sensor’s field of view. However, this error increased to a magnitude of 4.5° as the final orientation in the experiments involving the coordination of fuzzy behaviors. Although this increase is larger compared to the angular displacement experiments, it remains a highly acceptable value for human-following applications.

A key advantage of this architecture is its modularity and scalability. These design principles allow for easy adaptation and future enhancements, including the integration of additional sensors, new behaviors, or the application of the system to a broader range of assistive tasks. This flexibility makes it a highly adaptable solution for various personal assistance scenarios, such as aiding the elderly or individuals with disabilities.

Potential future directions for this work include refining the system’s ability to operate in more dynamic and unpredictable environments. Incorporating machine learning algorithms for human behavior prediction and trajectory planning could further improve the robot’s ability to manage complex interactions. Additionally, optimizing the computational load of the control architecture would enable its deployment on even more resource-constrained robotic platforms, broadening its applicability across diverse domains.

The development and validation of the Carlitos-Care platform represent a significant advancement in the field of assistive robotics. The system not only addresses critical challenges in human-following tasks but also provides a scalable and cost-effective solution for enhancing personal assistance technology. This research sets a strong foundation for future innovations in autonomous robots designed to improve the quality of life for individuals requiring assistance in their daily activities.

## Figures and Tables

**Figure 1 sensors-24-07796-f001:**
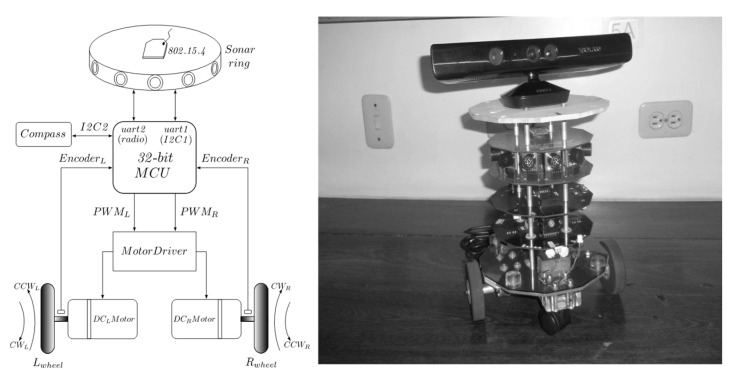
Block diagram representation and physical appearance of Carlitos-Care mobile robotic platform. Wireless communication is provided by an 802.15.4 XBee module. Source: Authors.

**Figure 2 sensors-24-07796-f002:**
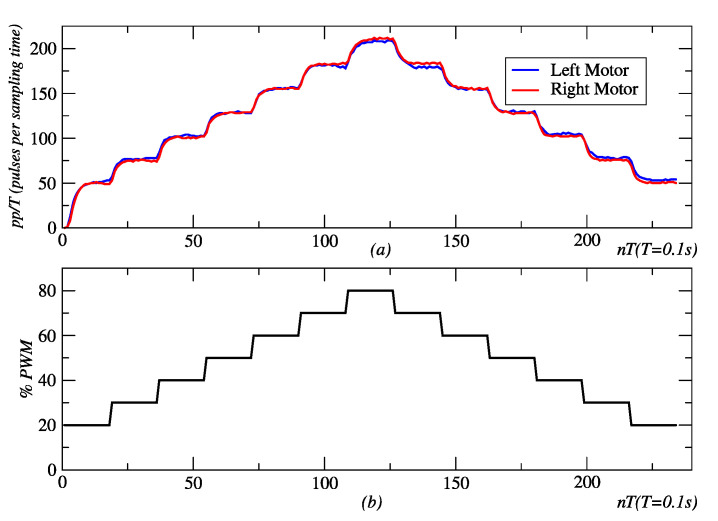
(**a**) robot effector response to a (**b**) time-variant duty-cycle PWM control signal (**b**). Source: Authors.

**Figure 3 sensors-24-07796-f003:**
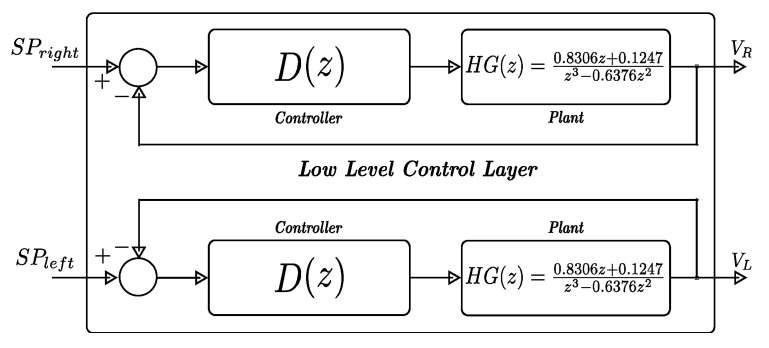
Low-level control layer implemented in the MCF51QE128-embedded system by means a couple of PI discrete-time controllers. Source: Authors.

**Figure 4 sensors-24-07796-f004:**
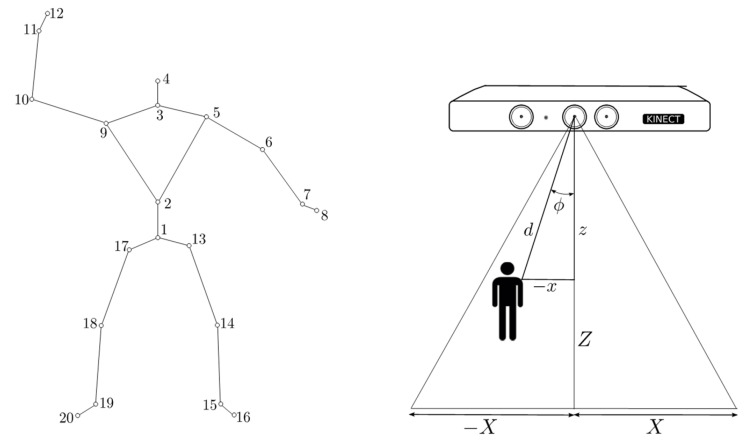
Skeletal joints and coordinate system of the Kinect sensor. For the follow-up of people, we isolate the articulation number 1. Source: Authors.

**Figure 5 sensors-24-07796-f005:**
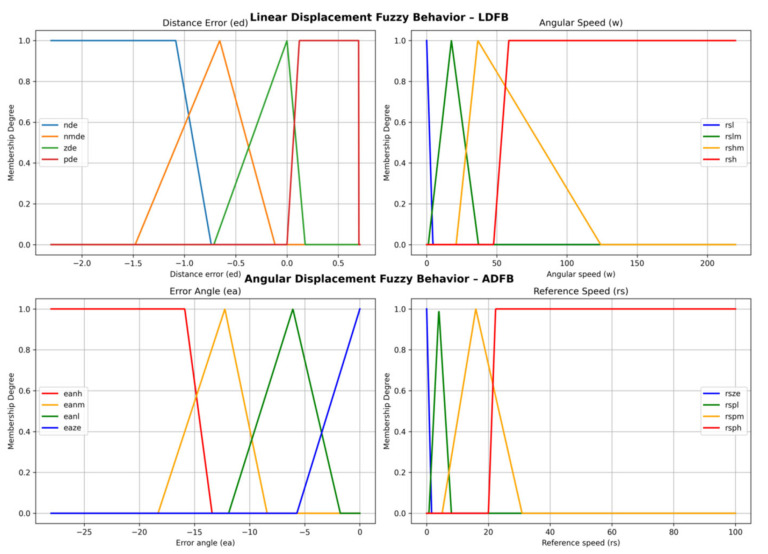
Fuzzy membership functions for LDFB and ADFB. Source: Authors.

**Figure 6 sensors-24-07796-f006:**
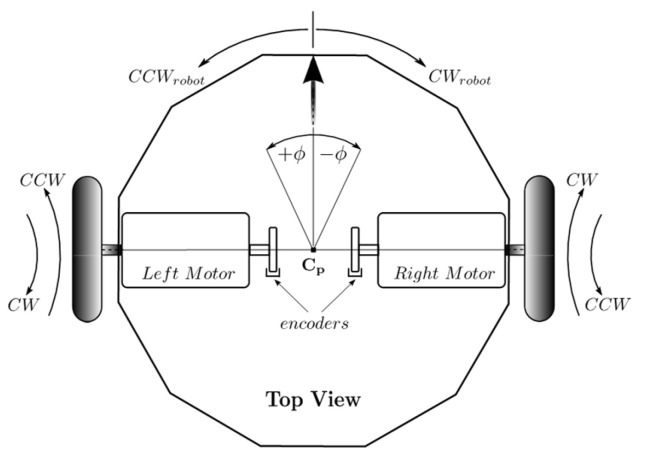
Control of movements in a differential robot. When the motor speeds are equal in magnitude, either pure linear or angular displacement can be achieved by controlling the direction of motor rotation. Source: Authors.

**Figure 7 sensors-24-07796-f007:**
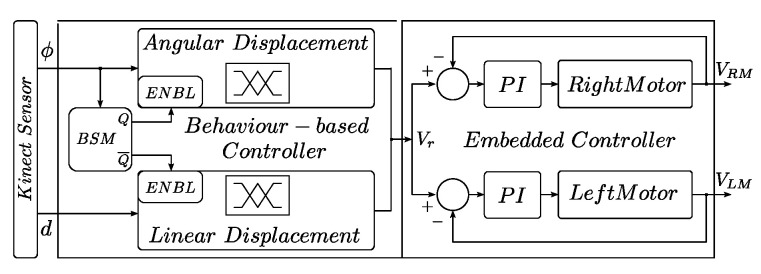
Proposed two-level control architecture BSM scheme. Source: Authors.

**Figure 8 sensors-24-07796-f008:**
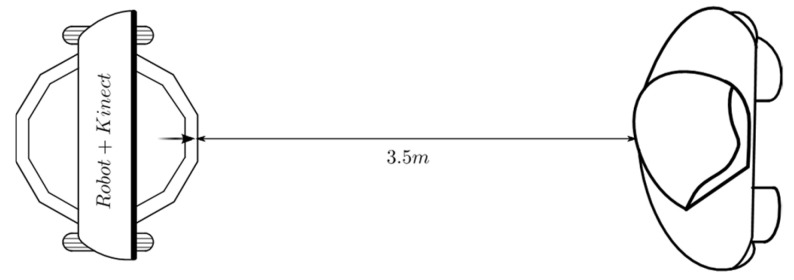
Linear displacement behavior: the robot approaches a person standing. Source: Authors.

**Figure 9 sensors-24-07796-f009:**
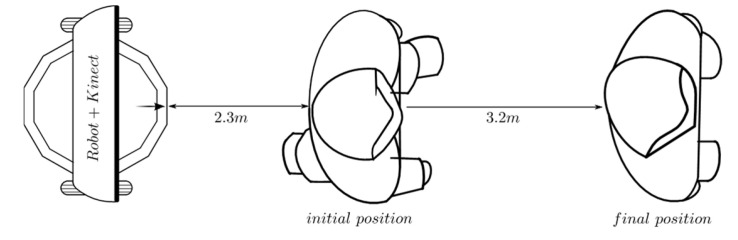
Linear displacement behavior: the robot approaches a person standing and follows them. Source: Authors.

**Figure 10 sensors-24-07796-f010:**
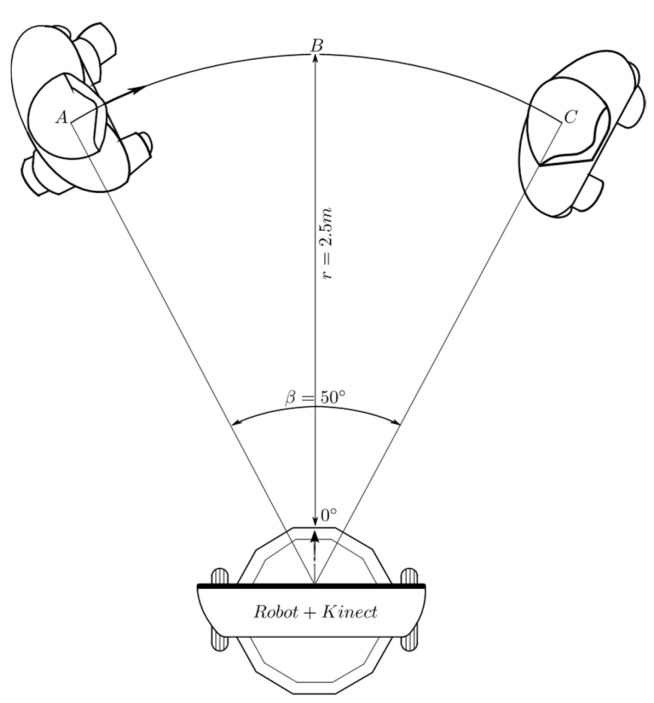
Angular displacement behavior: the robot makes an angular displacement (left to right) that maintains the person in the center of the scene. Source: Authors.

**Figure 11 sensors-24-07796-f011:**
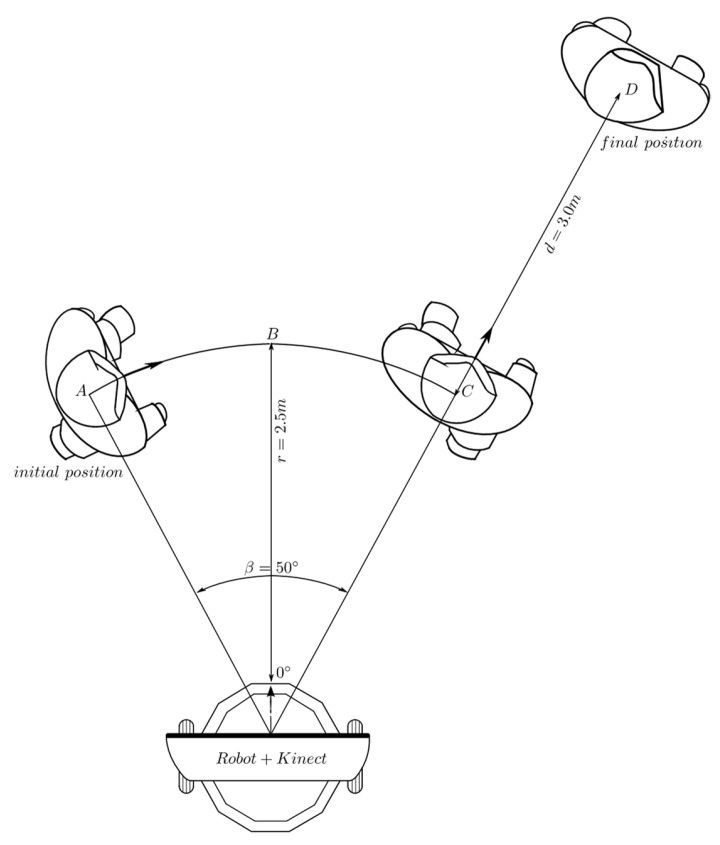
Behaviors coordination: the robot first makes an angular displacement and then a linear displacement. Source: Authors.

**Figure 12 sensors-24-07796-f012:**
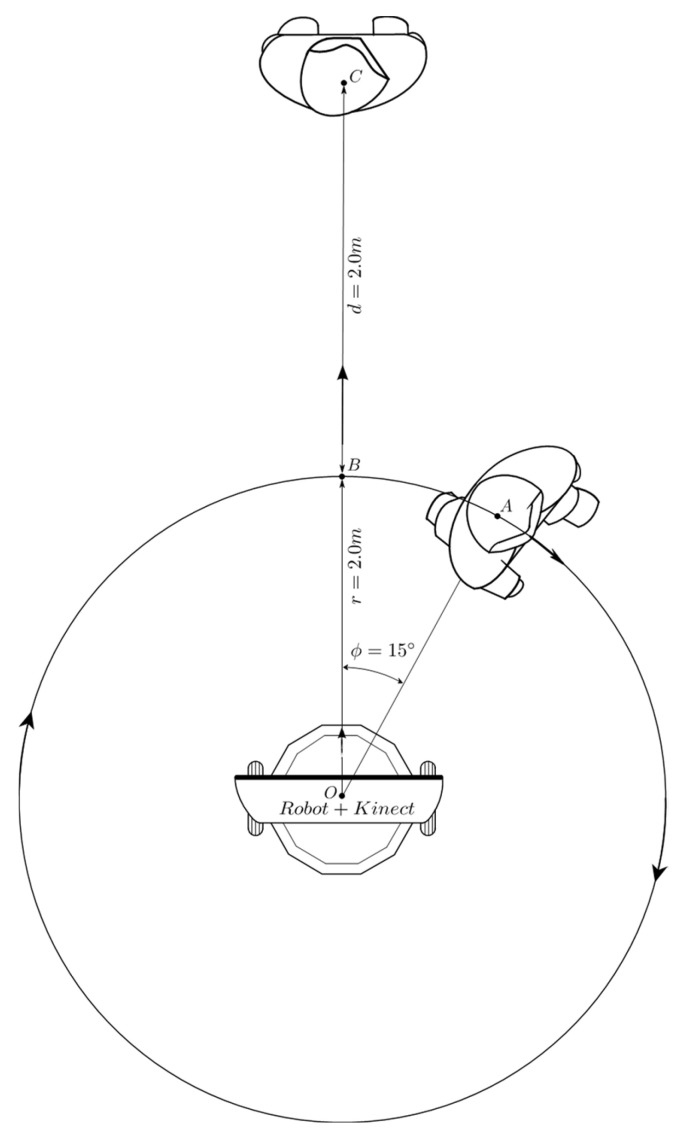
Complete circle displacement and linear displacement test. Source: Authors.

**Figure 13 sensors-24-07796-f013:**
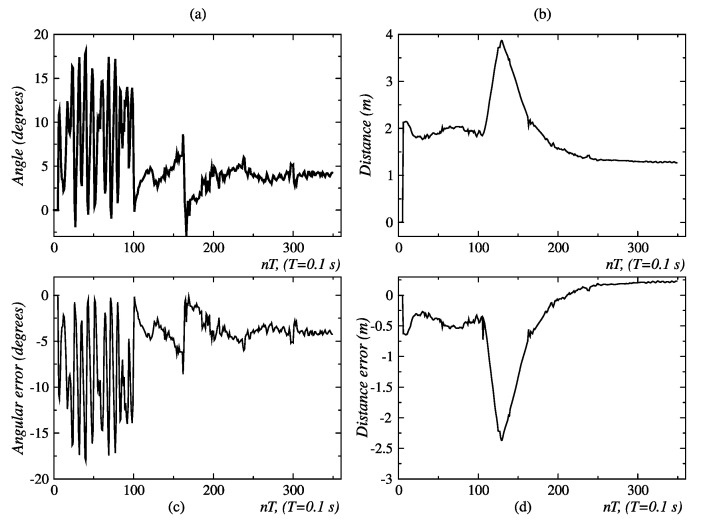
Angle distances and errors related to circular and linear displacement, detailing: (**a**) the angle between the person and the robot, (**b**) their separating distance, (**c**) the angular error (deviation from the desired orientation), and (**d**) the distance error (difference from the desired position). Source: Authors.

**Figure 14 sensors-24-07796-f014:**
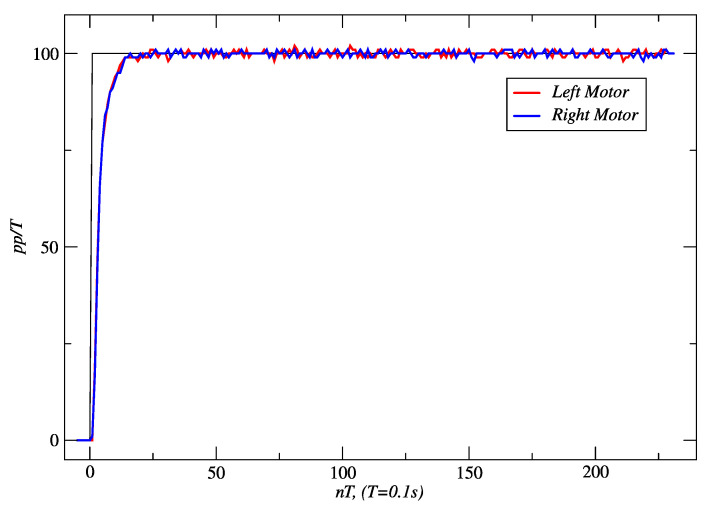
PPC responses of the right and left motors to a set-point of 100 *pp*/*T*. Source: Authors.

**Figure 15 sensors-24-07796-f015:**
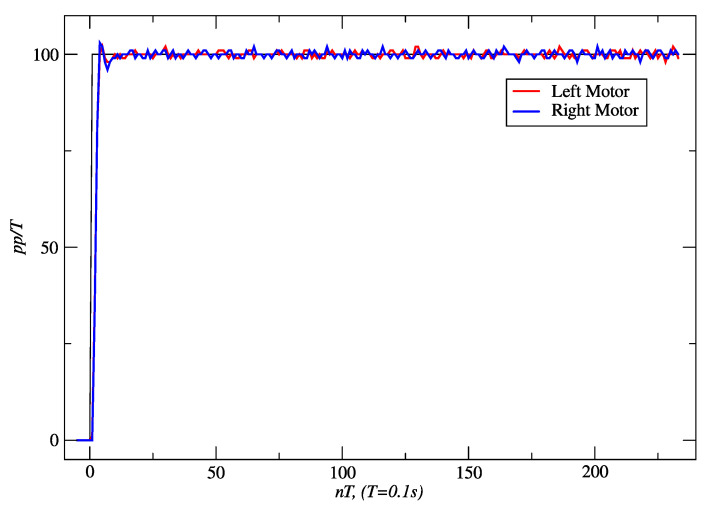
IMC responses of the right and left motors to a set-point of 100 *pp*/*T*. Source: Authors.

**Figure 16 sensors-24-07796-f016:**
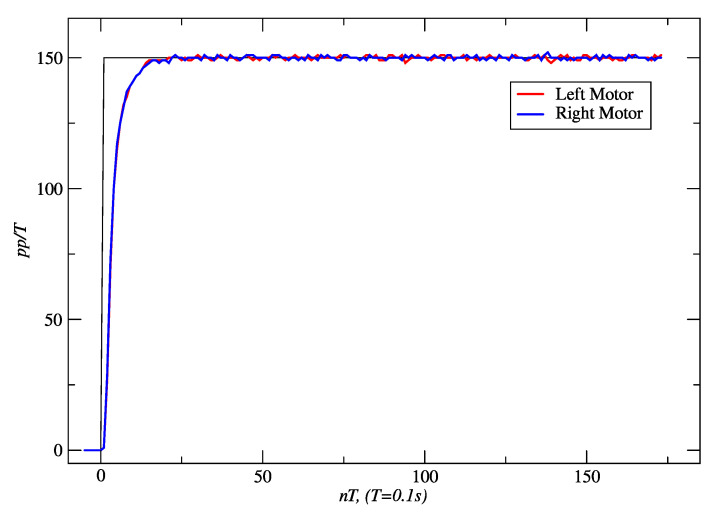
PPC responses of the right and left motors to a set-point of 150 *pp*/*T*. Source: Authors.

**Figure 17 sensors-24-07796-f017:**
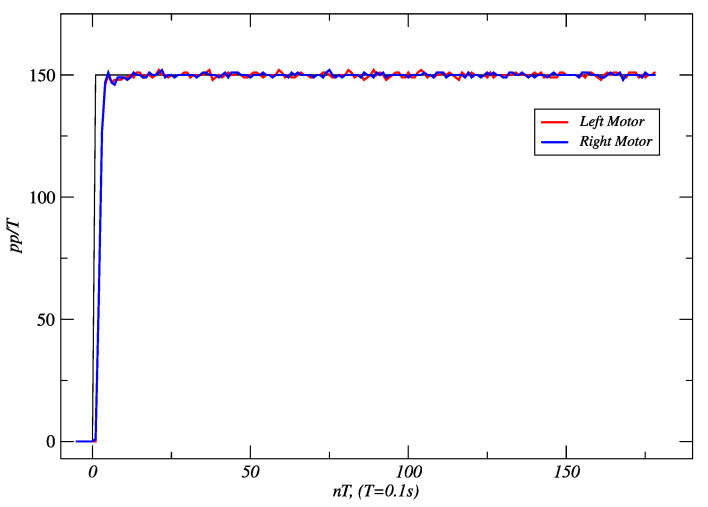
IMC responses of the right and left motors to a set-point of 150 *pp*/*T*. Source: Authors.

**Figure 18 sensors-24-07796-f018:**
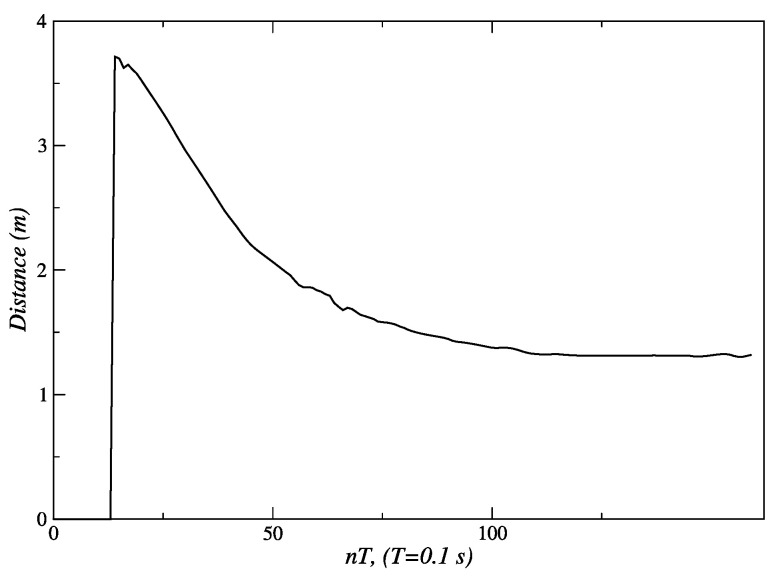
Distance measured by the Kinect during the linear displacement of the person. Source: Authors.

**Figure 19 sensors-24-07796-f019:**
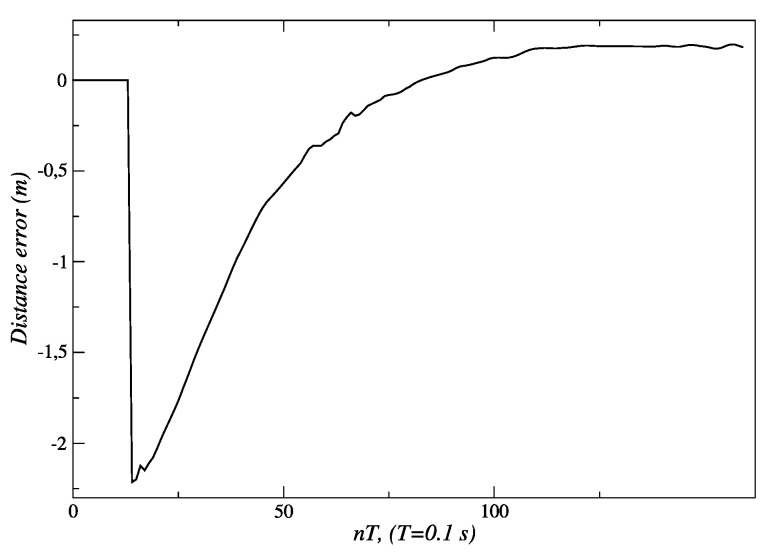
Distance error measured by the Kinect during the linear displacement of the person. Source: Authors.

**Figure 20 sensors-24-07796-f020:**
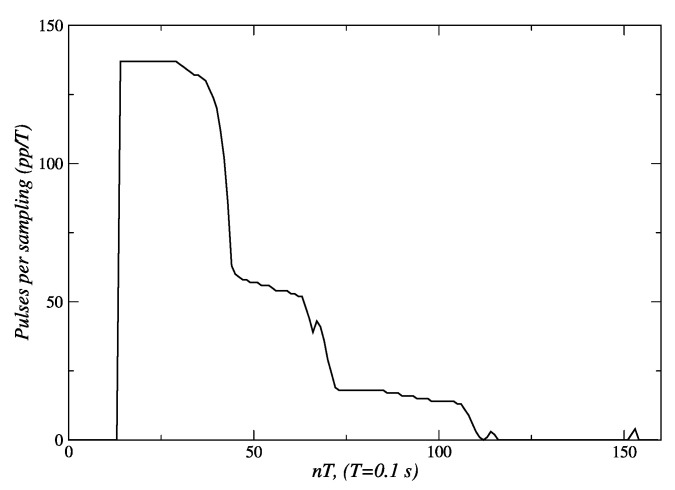
Velocity references sent to the actuators of the robot. Source: Authors.

**Figure 21 sensors-24-07796-f021:**
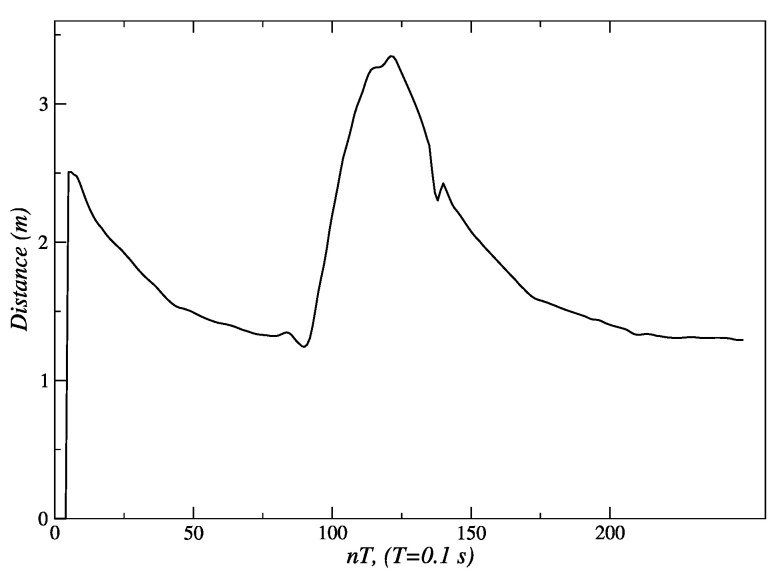
Distance between the robot and a person initially standing and then being followed. Source: Authors.

**Figure 22 sensors-24-07796-f022:**
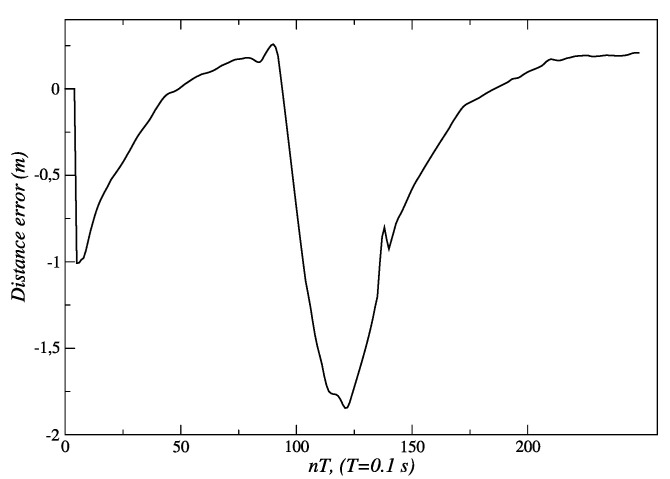
Distance error between the robot and a person initially standing and then being followed. Source: Authors.

**Figure 23 sensors-24-07796-f023:**
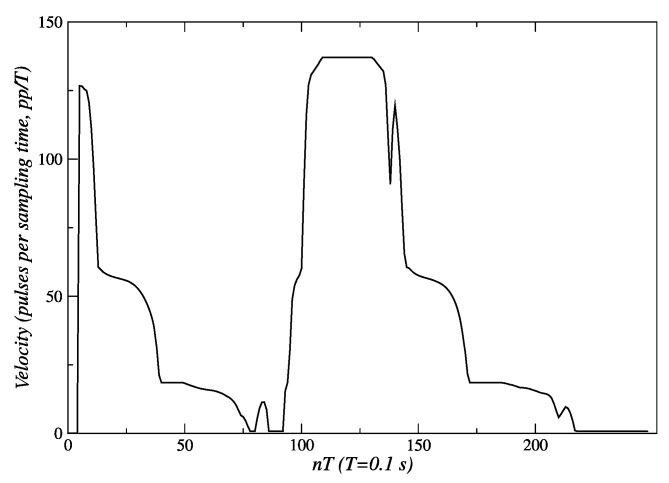
Velocity set-points are sent to the actuators of the robot for the linear following of persons. Source: Authors.

**Figure 24 sensors-24-07796-f024:**
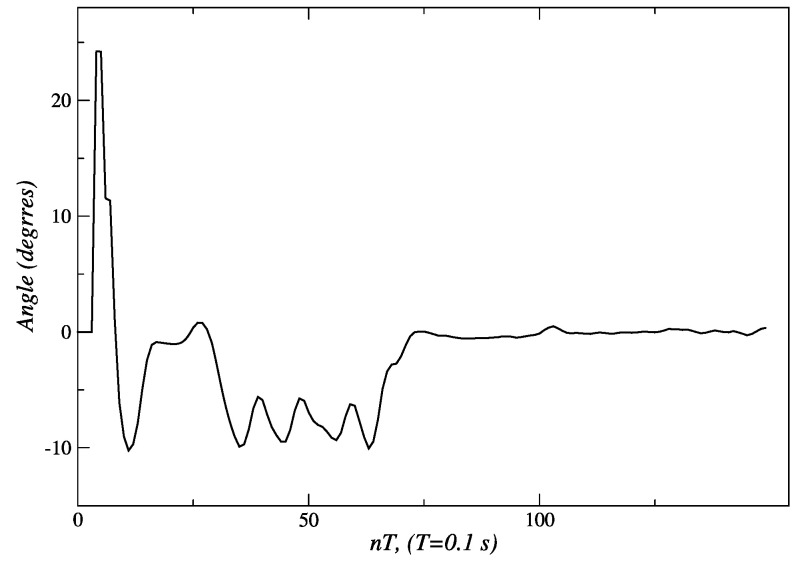
Angular displacement behavior: the robot makes an angular displacement (right to left) that maintains the person in the center of the scene. Source: Authors.

**Figure 25 sensors-24-07796-f025:**
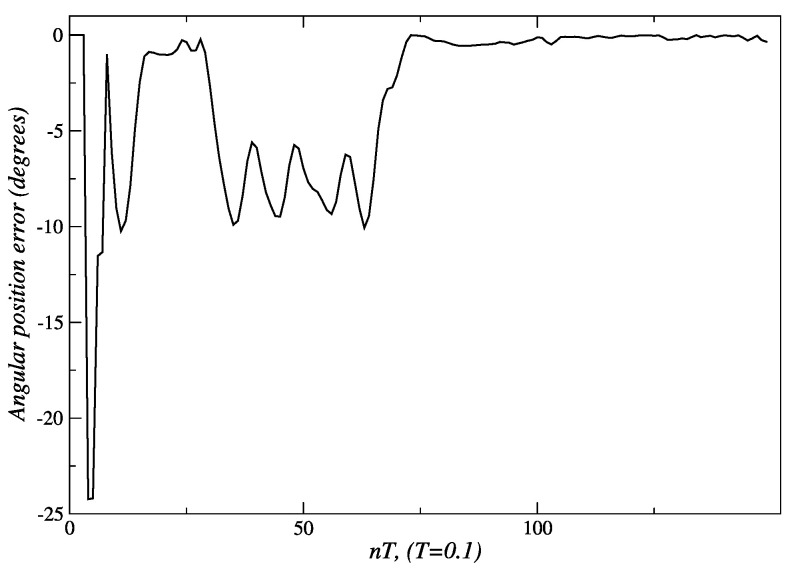
Angular displacement error: the robot makes an angular displacement (right to left) that maintains the person in the center of the scene. Source: Authors.

**Figure 26 sensors-24-07796-f026:**
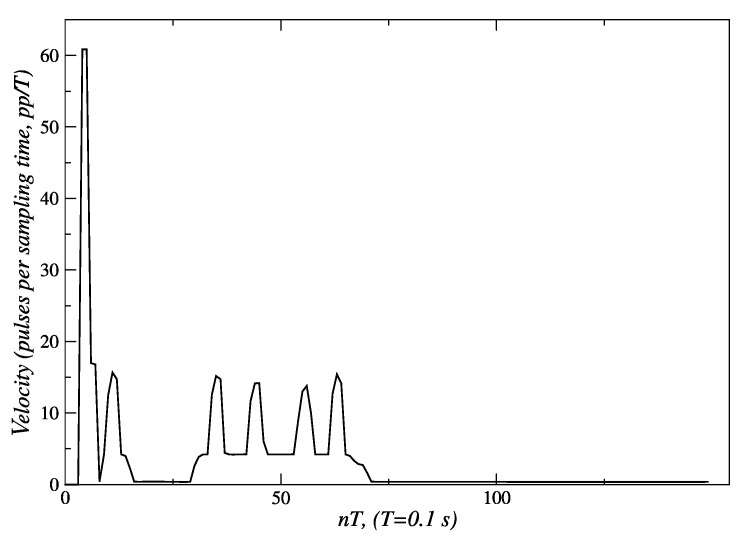
Velocity angular behavior: the robot makes an angular displacement (right to left) that maintains the person in the center of the scene. Source: Authors.

**Figure 27 sensors-24-07796-f027:**
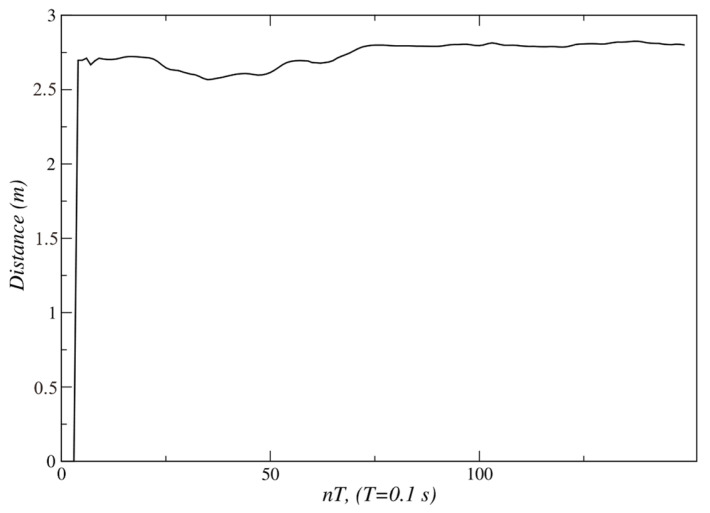
Distance to the person during the angular displacement. Source: Authors.

**Figure 28 sensors-24-07796-f028:**
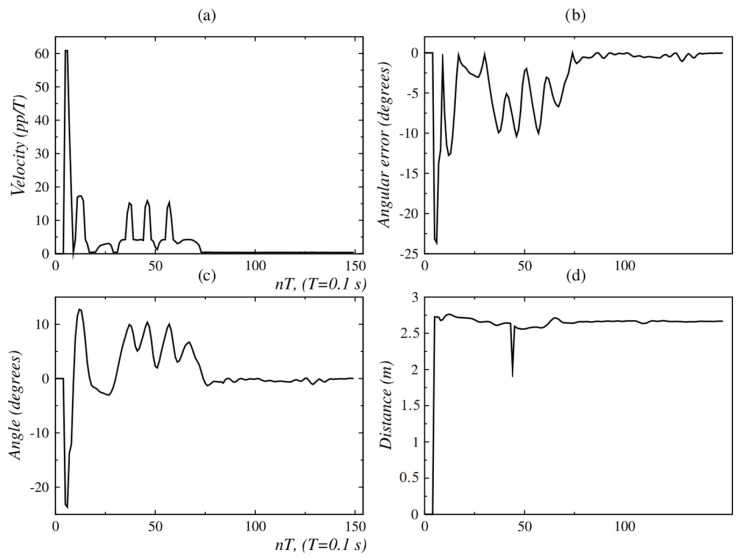
Angular displacement behavior: the robot makes an angular displacement (right to left) that maintains the person in the center of the scene, showing: (**a**) speed, (**b**) angular error, (**c**) angle, and (**d**) distance. Source: Authors.

**Figure 29 sensors-24-07796-f029:**
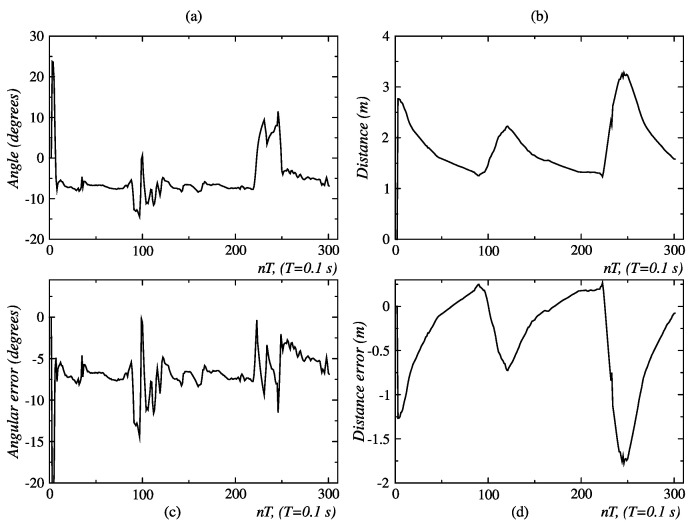
Angles, distances, and errors related to the arc of the person walking away, detailing: (**a**) angles, (**b**) distance, (**c**) angular error, and (**d**) distance error. Source: Authors.

**Figure 30 sensors-24-07796-f030:**
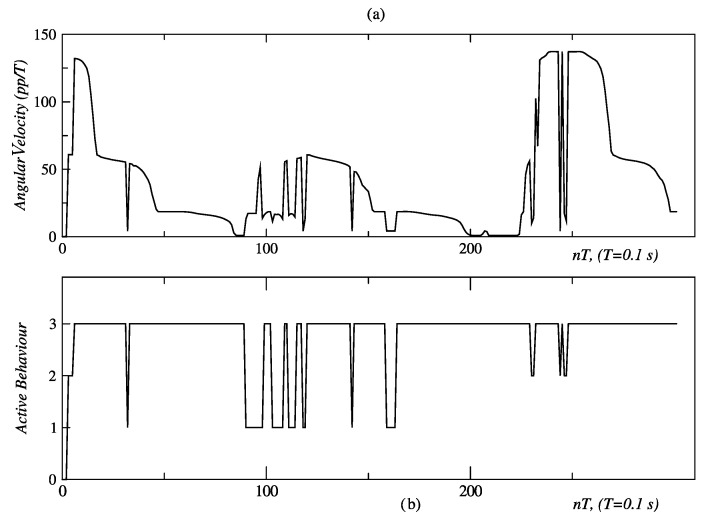
Angular velocity and active behavior related to the arc of the person walking away, showing: (**a**) angular velocity, and (**b**) active behavior. Source: Authors.

**Figure 31 sensors-24-07796-f031:**
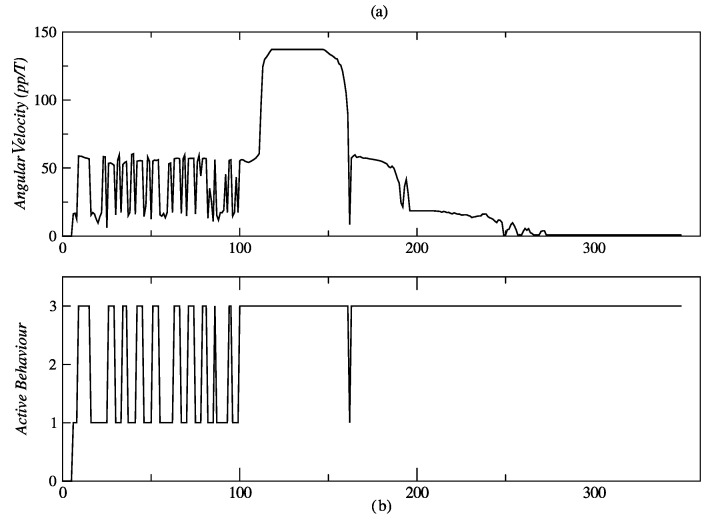
Angular velocity and active behavior related to circular and linear displacement, showing: (**a**) angular velocity, and (**b**) active behavior. Source: Authors.

**Table 1 sensors-24-07796-t001:** Variables associated with FIS controllers. Source: Authors.

Behavior	Input Variable	Output Variable
Linear displacement	*e_d_*: distance error between person and robot	Speed references for IMC
Angular displacement	*e_a_*: angular error between person and robot	Speed references for IMC

**Table 2 sensors-24-07796-t002:** LDFB characterization. Source: Authors.

Input: Distance Error (*e_d_*)	Output: Angular Speed of Wheels (*ω*)
U = {*e_d_* ∈ ℝ | −2.3 ≤ *e_d_* ≤ 0.7} T(*e_d_*) = {*nde*, *nmde*, *zde*, *pde*} G: Prefixes: *n: negative, m: medium* z: zero, p: positive Suffixes: *de: distance error*	V = {*ω* ∈ ℝ | 0 ≤ *ω* ≤ 220} T(*ω*) = {*rsl, rslm, rshm, rsh*} G: Prefixes: *rs: references of speed* Suffixes: *l: low, m: medium* *h: high*
M: *nde → trapmf [−2.3, −2.3, −1.085, −0.739]* *nmde → trimf [−1.48, −0.6558, −0.117]* *zde → trimf [−0.713, 0, 0.177]* *pde → trapmf [0, 0.12, 0.7, 0.7]*	M: *rsl → trimf [0, 0, 4.47]* *rslm → trimf [1.18, 17.63, 36.9]* *rshm → trimf [21, 36.43, 124]* *rsh → trapmf [47.7, 58.53, 220, 220]*

**Table 3 sensors-24-07796-t003:** Driving of motors for rotating the robot. Source: Authors.

Left Motor	Right Motor	Robot Displacement
CCW	CW	Linear forward
CW	CCW	Linear back
CW	CW	CCW
CCW	CCW	CW

**Table 4 sensors-24-07796-t004:** ADFB characterization. Source: Authors.

Input: Error Angle (*e_a_*)	Output: Reference Velocity for Angular Displacement (v*_ra_*)
U = {*e_a_* ∈ ℝ | −28 ≤ *e_a_* ≤ 0} T(*e_a_*) = {*eanh, eanm, eanl, eaze*} G: Prefixes: *ea: error angle* Suffixes: *n: negative, m: medium* *ze: zero, h: high, l: low*	V = {*v_ra_* ∈ ℝ | 0 ≤ *v_ra_* ≤ 100} T(*v_ra_*) = {*rsze, rspl, rspm, rsph*} G: Prefixes: *rs: references of speed* Suffixes: *l: low, m: medium* *h: high, p: positive, ze: zero*
M: *eanh → trapmf [−28, −28, −15.88, −13.4]* *eanm → trimf [−18.3, −12.24, −8.41]* *eanl → trimf [−11.9, −6.073, −1.77]* *eaze → trapmf [−5.71, 0.0299, 0, 0]*	M: *rsze → trimf [0, 0.534, 1.603]* *rspl → trimf [0.748, 3.95, 8.013]* *rspm → trimf [5.02, 15.92, 30.9]* *rsph → trapmf [20, 22.33, 100, 100]*

**Table 5 sensors-24-07796-t005:** Performance indexes of PPC and IMC. Source: Authors.

	Left IAE	Motor ISE	Right IAE	Motor ISE
Pole Placement	553	48,955	549	47,855
Internal Model	303	31,561	301	31,443

## Data Availability

The data presented in this study are available upon request from the corresponding authors.

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
