# Peer review of "Lightweight Two-Layer Control Architecture for Human-Following Robot"

_sensors, 2024, doi:10.3390/s24237796_

Round 1

Reviewer 1 Report

Comments and Suggestions for Authors

This study proposes a robotic platform for human tracking. Two embedded controllers, pole placement and internal model control, were implemented and compared. Furthermore, a fuzzy behavior based control system was implemented using a sensor to capture distance and angular data to estimate the speed on the robot’s motors. Additionally, three types of experiments were conducted to assess the robot’s performance: linear approximation and tracking of a person, angular approximation and tracking of a person, and a combination of angular and linear displacements.

Please find below some comments and suggestions.

Abstract:

It would be convenient to provide numerical results at “(3) Results” in order to support the sentences (lines 18-21):  “Experimental validation demonstrated that the proposed architecture enables the robot to follow a person in real-time, maintaining a safe distance even under occlusion conditions caused by obstacles and people circulation. The IMC-based controller exhibited superior performance in maintaining trajectory accuracy with minimal error. ” For instance, please specify the distance that is maintained instead of using “safe distance”. Similarly, please provide numerical data regarding: “superior performance”, and “minimal error”.

Introduction:

Please re-write the sentence of lines 37-38 as follows: “A broad overview of potential uses of service robots to assist the elderly and disabled people is presented in [8].” Please rewrite the sentence of lines 38-41 in a similar way.

Please improve the writing of the ending of the following sentence (lines 47-48): “This robot incorporates an RGB-D sensor used only for object classification and manipulation, but not for movement tracking of people.”

Materials and methods:

Please specify how many tests were performed instead of using “multiple tests” (line 217)

Please specify the meanings of kT in equations 29 and 30

It would be convenient to use a different type of lines for each motor in Figures 2, 3, 4, 5, and 6, so that left and right motors can be distinguished.

This section could be organized as follows:

2.1. Mobile Robotic Platform

2.2. Effectors Modelling and Control

2.2.1. Pole Placement Method

2.2.2. Internal Model

2.3. Linear and Angular Displacements

2.3.1 Linear Displacement Fuzzy Behavior – LDFB

2.3.2. Angular Displacement Fuzzy Behavior – ADFB

 2.4. Experimental Designs

2.4.1. Experimental Design for Embedded Controllers

2.4.2. Experimental Design for Fuzzy Behaviors

2.4.3. Experimental Design for Linear Tracking

2.4.4. Experimental Design for Angular Tracking

2.4.5. Experimental Design for Behavior Coordination

Results and Discussions:

In this section, it is recommended to explain only the output of the experiments. Details regarding how the proposal was implemented, and how the experiments were conducted should be in the method section.

This section presents details that are related to the methods used to test the authors’ proposal. Consequently, these parts should be moved to the method section. For instance, the subsections “linear displacement”, “angular displacement”, “linear displacement fuzzy behavior”, “angular displacement fuzzy behavior”,etc., explain details regarding the methods used to test the authors’ proposal rather than experimental results .

Regarding the FIS, it would be convenient to create figures in which the input fuzzy sets and output fuzzy sets are described. This should be moved to method section

Additionally, it is highly recommended to conduct a statistical analysis using either parametric tests (e.g., t-test) or non-parametric tests (e.g., Wilcoxon signed rank test) depending on the samples, so that it can be determined whether there is significant difference between the pole placement and internal mode controllers in the metrics used to assess their performances.  

Please revise if “φmin” should be duplicated (line 291)

Please try to write using neutral gender in lines 295-296. Try using plural nouns (i.e., people and them)

This section could be organized as follows:

3. Results

3.1. Results for Embedded Controllers

3.2. Statistical Analysis of Embedded Controllers

3.3. Results for Fuzzy Behaviors

3.4. Results for Linear Tracking

3.5. Results for Angular Tracking

3.6. Results for Behavior Coordination

Although the results and discussion could be merged into one section, it would be convenient to present the discussion as a separate section.

4. Discussion

In this section, the key findings could be emphasized. Moreover, a comparison with previous studies should be presented. It would be recommended to compare the results of the authors’ proposal with results of previous studies. Furthermore, please mention the limitations that the proposal might have.

Conclusions:

It would be convenient to provide numerical results to support the conclusions.

Comments on the Quality of English Language

It would be convenient to perform a proof-reading. There are several sentences that need to be rewritten.

Author Response

Respected reviewer. In response to the recommendations made to the article, the authors inform you that we have attended to all of them and have included them in the manuscript.
Best regards.

Reviewer 2 Report

Comments and Suggestions for Authors

This paper proposes the design and implementation of a lightweight two-layer control architecture for a human-following robot. Experimental validation shows that the proposed architecture enables the robot to follow humans in real time and maintain a safe distance even under occlusion conditions. The IMC-based controller excels in maintaining trajectory accuracy and minimum error. The paper is easy to follow. I recommend this paper to be accepted after revision.

More detailed comments are listed as follows: 

Authors proposed a lightweight two-layer control architecture. How to define lightweight? Compared to what?

Comments on the Quality of English Language

More detailed comments are listed as follows: 

Authors proposed a lightweight two-layer control architecture. How to define lightweight? Compared to what?

Author Response

Respected reviewer. In response to the recommendations made to the article, the authors inform you that we have attended to all of them and have included them in the manuscript.
Best regards,

Round 2

Reviewer 1 Report

Comments and Suggestions for Authors

Although the authors did not perform the suggested statistical analysis as they mentioned that this will done as future work, the remaining comments were fulfilled.